# Heat Treatment of Seeds to Control Invasive Common Ragweed (*Ambrosia artemisiifolia*), Narrow-Leaved Ragwort (*Senecio inaequidens*) and Giant Hogweed (*Heracleum mantegazzianum*)

**DOI:** 10.3390/plants13030341

**Published:** 2024-01-23

**Authors:** Rea Maria Hall, Bernhard Urban, Nora Durec, Katharina Renner-Martin, Hans-Peter Kaul, Helmut Wagentristl, Gerhard Karrer

**Affiliations:** 1Institute of Agronomy, University of Natural Resources and Life Science, Vienna, 3430 Tulln an der Donau, Austriahans-peter.kaul@boku.ac.at (H.-P.K.); 2Institute of Botany, University of Natural Resources and Life Science, Vienna, 1180 Vienna, Austria; gerhard.karrer@boku.ac.at; 3Institute of Mathematics, University of Natural Resources and Life Science, Vienna, 1180 Vienna, Austria; 4Experimental Farm, University of Natural Resources and Life Science, Vienna, 2301 Groß-Enzersdorf, Austria; helmut.wagentristl@boku.ac.at

**Keywords:** hot air, hot water, hot steam, hot foam, seed intentional deterioration, invasive alien species

## Abstract

The intended or unintentional transport of soil material contaminated with weed seeds is one of the most important drivers in the spreading dynamics of invasive alien plants (IAPs). This phenomenon can be observed at any kind of construction site. Typical transfer of soil contaminated with IAP seeds can be observed along with road construction (soil translocation) or road maintenance services (deposit of mown plant biomass). Thus, an effective inactivation of these seeds by heating can avoid the spread of IAPs substantially. In the present study, the effects of various thermal control techniques (dry air heating and wet heating with hot steam, hot water, and hot foam) on seed survival of the widespread European IAPs common ragweed (*Ambrosia artemisiifolia*), narrow-leaved ragwort (*Senecio inaequidens*), and giant hogweed (*Heracleum mantegazzianum*) are discussed. Dry and wet seeds which were either uncovered or covered with soil were tested for survival at different treatment temperatures and different exposure times. Results revealed that particularly dry seeds of all three species could withstand temperatures of 100 °C for at least 6 h in climate chambers. Dry seeds of common ragweed and narrow-leaved ragwort survived exposure times of up to 48 h. Wet seeds were significantly more susceptible to heat treatments. Giant hogweed seeds were completely killed after 12 h at 70 °C. The exposure of IAP seeds to hot water was generally more effective than the treatment with hot steam. The treatment with hot foam was only effective when seeds were lying unprotected on the soil surface. Dry seeds of all the three species survived hot foam application in the field when they were covered by vegetation and leaf litter or soil. Due to the robustness of the seeds, a preventive management of IAPs by an efficient control before seeds formation is substantial to avoid their further dispersal.

## 1. Introduction

Transcontinental spreading of invasive alien plants (IAP) by seeds increased significantly during the last two centuries [1]. Spreading processes and seed traits of IAPs in the non-native ranges are basically the same as in the native range [2]. Once the introduced plants arrive at non-native regions, they often quite easily established and naturalized [3,4]. Subsequent invasiveness is mostly bound to the dispersal traits of seeds [5]. Seed dispersal by wind, water or animals are highly associated with specific morphological traits like winged seeds in several Apiaceae or umbrellas in many Asteraceae [6]. Seeds lacking specific adaptations are commonly spread accidentally, i.e. when incorporated in transported soil or crops. Dispersal success of IAPs in a geographical area is positively associated with high numbers of released seeds, and dispersal in time (seed survival) by dormancy in soil seed banks [7]. The intentional and/or unintentional transport of soil material contaminated with weed seeds is one of the most important drivers in the spreading dynamics of IAPs [8]. This type of unintended transportation of IAP seeds can be observed specifically in various types of construction sites. At the regional scale, typical movements of contaminated soil happen along with road construction (i.e., soil translocation [9]) or road maintenance services (i.e., deposit of mown plant biomass [10]). This is most evident for common ragweed (*Ambrosia artemisiifolia*, [11,12,13]) that endangers human populations by its allergic pollen [14,15].

At the continental scale, seeds of IAPs can also be spread by infested crop seed material [12,13,16,17]. One big driver of distribution is transcontinentally traded goods that reach Europe via big harbors [16]. This is assumed to be the starting point of the invasion of the highly poisonous narrow-leaved ragwort (*Senecio inaequidens*) in Europe [18]. It produces high numbers of light wind-dispersed seeds that can also be integrated into the persistent soil seed bank [19,20]. From the beginning of the 20th century onwards, intended planting and seed trade for gardening served as a source of IAP seed dispersal [21]. The IAP giant hogweed (*Heracleum mantegazzianum*) belongs to these garden escapes and causes severe skin injuries of humans and is also rapidly spreading throughout Europe [22,23].

The three mentioned species belong either to the European list of Invasive Alien Species of Union concern (Commission Implementing Regulation (EU) 2022/1203 of 12 July 2022 amending Implementing Regulation (EU) 2016/1141 to update the list of invasive alien species of Union concern) or to the EPPO list of invasive alien species (https://www.eppo.int/ACTIVITIES/invasive_alien_plants/iap_lists#iap; accessed on 10 January 2024). All three species cause harm to humans and are to be controlled in many European countries [13,23,24]. Furthermore, these species are focus species for control measures along the roadsides in many European countries [25,26].

Seeds are often targeted as a life cycle stage to focus on when control options of IAPs are to be developed. Options include harvesting seeds that stick on plants directly along with biomass reduction (cutting) or making seeds unviable in soil moved in the course of construction work. Heating seeds can kill living embryos and avoid germination at interim storage or the final destination of material that contains IAP seeds. Because thermal weed control is expensive and laborious, there is a need to know the minimum temperature and treatment duration requirements for inactivating seeds. Due to increasing restrictions in the use of herbicides, weeding without pesticides has become more and more popular in the recent years. New technologies like weed treatment with hot water, steam or hot foam are nowadays increasingly common in various sectors like agriculture, road maintenance, landscaping, or municipal plant management [27,28,29,30]. Living green tissue is destroyed by these methods. Corresponding effects on seeds are not well documented. Therefore, a project was initiated by the University of Natural Resources and Life Sciences, Vienna in cooperation with the Bavarian Road maintenance services to study heat exposure effects on seeds of widespread invasive weeds. Three well known invasive species that are widespread, but challenging to control [20,23,31] were selected (common ragweed, narrow-leaved ragwort, and giant hogweed) to test the effects of various thermal control techniques on the survival rates of seeds. Therefore, laboratory trials were implemented in which wet and dry seeds of these three species were tested under hot air treatment at five temperature levels (60–100 °C) and seven time intervals (0.5–48 h). Thereby, seeds were either protected (incorporated in soil) or unprotected. In addition, we conducted tests on hot steam and hot water treatments on wet and dry seeds of the three species in six time intervals (0.5–40 min) as well as hot foam treatment. The aim of this study was to (1) investigate the effectiveness of the treatment techniques (hot-air treatment, hot moisture treatment), and (2) determine the necessary temperature requirements and treatment duration to fully mortify seeds of these IAPs. Based on the results of this study, the efficacy of weed control measures may be enhanced.

## 2. Results

The study focusses on effects of heat treatments to kill IAP seeds. Therefore, the main focus is on the specific heating methods and their effect on the three different species.

### 2.1. Heat Treatment

***Common ragweed:*** Prior to the treatment, the germination rate of the dry, untreated ragweed seeds (control) was 98.2%, and those of the untreated wet seeds came up to 98.0%. Treatment temperatures of 60 °C and 70 °C had nearly no effect on the viability of dry seeds. After 48 h, the survival rate of unprotected dry seeds was still 97% when treated with 60 °C, and 89.9% at 70 °C. Protected dry seeds showed a survival rate of 95.8% at 60 °C and 73.2% at 70 °C (Figure 1a–d; Appendix A).

Wet seeds were slightly more vulnerable, as their survival rate at 60 °C showed a constant decrease after 6 h of exposure when they were unprotected. After 12 h of treatment, their average survival rate attained 45.2% on average. At 70 °C, the first drop in the survival rate was observed after 3 h, when already more than half of the seeds were dead. Longer treatment duration (6–48 h) reduced survival rate by almost 90%, but even after 48 h of exposure, 13.5% of the wet seeds survived. However, when the wet seeds were protected, they only showed a slight loss in viability of 15.5% after 48 h of exposure to 60 °C. At 70 °C, 64.2% of the wet seeds survived the heat treatment for 48 h.

The first complete killing was observed with unprotected wet seeds exposed to 80 °C for 24 h. The hot air treatment duration necessary for 0% survival was reduced to 12 h when wet, unprotected seeds were exposed to 90 °C and 100 °C, respectively, even though the survival rate decreased sharply (approx. 95%) already after 1 h of exposure to these high temperatures. When wet seeds were incorporated in soil the treatment with 80 °C and 90 °C did not totally kill the seeds. Protected wet seeds had to be treated for at least 12 h at 100 °C to fully die off.

***Narrow-leaved ragwort:*** As summarized in Appendix A), seeds of narrow-leaved ragwort were slightly more vulnerable to the heat treatment compared to seeds of common ragweed, even though the treatment with 60 °C did not show any effects, irrespective of the seeds being wet or dry, protected, or unprotected (Figure 2a–d). After 48 h of exposure to 60 °C, all seed still showed a mean survival rate of 87.7% which did not differ from the untreated control seeds (average germination rate of 94.5% with dry seeds, and 94.0% with wet seeds). At 70 °C, only wet seeds showed a decrease in their survival rate of 44% when they were unprotected and 20.8% when protected by soil. As with ragweed, a full killing of seeds was only achieved with wet seeds. Unprotected, wet seeds fully died off at 80 °C already after 0.5 h of exposure. The same treatment time was observed with protected, wet seeds but an exposure to 90 °C was necessary. At 80 °C, protected, wet seeds showed an average decrease in the survival rate of 67%, independent of the treatment time. In contrast, dry seeds proved themselves as very robust as 22% of the unprotected seeds survived a heat treatment with 100 °C for 48 h. When they were incorporated in soil, as many as 39% of the seeds were viable after this treatment.

***Giant hogweed:*** Giant hogweed seeds were most vulnerable to heat treatments, compared to the other two species. Both untreated, dry and wet seeds showed an average germination rate of 86%. A treatment temperature of 60 °C had almost no effect on dry and wet seeds during 6 h of exposure. A decline of approx. 20–30% in the survival rate was noticeable after 12 h of exposure, irrespective of the seeds being protected or unprotected (Figure 3a–d; Appendix A). Particularly, with dry seeds the same minor effect of a 20–30% reduction was observed at even 70 °C. However, this temperature was enough to fully kill off unprotected wet seeds after an exposure time of 12 h. Even though more than half of the unprotected, wet seeds survived an exposure to 70 °C for 3 h, already after 6 h of exposure, only 4.1% of the seeds were viable. Also, wet seeds incorporated in soil showed a sharp decline in the survival rate of approx. 70% after 3 h of treatment at 70 °C. After 48 h only 5.1% of these seeds were still viable. Protected and unprotected wet seeds survived a treatment temperature of 80 °C for 1 h. After 3 h of exposure all of the seeds were dead. At 90 °C only 13% of the unprotected wet seeds survived a treatment for 0.5 h. When seeds were protected, they needed at least 3 h of treatment to fully die off. None of the unprotected wet seeds survived the treatment at 100 °C, independent of the exposure time, whereas 76.0% of the protected wet seeds were able to withstand 100 °C for at least 0.5 h. Dry seeds were a little bit more robust: For complete killing of seeds a treatment temperature of 90 °C for 48 h was necessary for unprotected and protected seeds, even though the survival rate declined by over 90% already after 24 h of exposure. At 100 °C a treatment time of 12 h was enough to fully kill off the seeds, irrespective of them being protected or unprotected.

The results of multiple regression analysis of experimental factors (Table 1) show that seed survival of common ragweed was significantly affected by all factors whereas seeds of narrow-leaved ragwort were not significantly affected by treatment time and seed protection state. In the case of giant hogweed, all factors except seed protection state turned out to influence seed survival significantly. These results were supported by one-way-ANOVA (Appendix A).

The GLMM analysis starts with the calculation of the explanatory power of each stand-alone factor. It is evident from Table 2 that particularly with common ragweed and giant hogweed the factor temperature had a high explanatory power. For the survival rate of narrow-leaved ragwort seeds it was crucial if they were dry or wet. However, when calculating the models, it became obvious that seed survival of all three species was mainly affected by a combination of the three factors: temperature, treatment time and moisture state. Even though the viability of common ragweed seeds varied significantly between protected and unprotected seeds, this factor played just a subordinated role when calculating the interaction and additive effects in GLMM. Particularly with narrow-leaved ragwort, the most parsimonious model for explaining seed survival was the interaction between temperature and moisture state, but the AICc value to the second-best model (temperature × moisture state + treatment time) did not exceed a Δi = 2, indicating that the treatment time has an additional explanatory value when combining it with the other factors, even though in multiple regression it was shown to be not significant. 

### 2.2. Dose–Response Curves

When testing the impact of temperature on seed survival, a non-linear response was found for the decline in the survival rate of all three species. The best fit for seed survival loss of all species was observed using Weibull model adjustment type 1 or type 2 with three parameters. To characterize the seed survival curves, we used the three factors moisture state (wet and dry), protection state (protected and unprotected) and temperature, defined as the temperature required to kill 99.9% (ED99) of the seeds, irrespective of the factor time (Figure 4a–c). According to the dose–response model, the most robust seeds are dry, protected seeds of narrow-leaved ragwort which require temperatures above 500 °C to die off immediately, followed by dry, protected seeds of common ragweed with an ED99 of 257.4 °C. Lower temperature requirements were calculated for wet, unprotected seeds. The ED99 of wet, unprotected seeds of common ragweed was calculated at 108.3 °C, for narrow-leaved ragwort at 76.8 °C and giant hogweed at 95.4 °C. As soon as these wet seeds are incorporated in soil, the ED99 rises up to 175.6 °C for common ragweed, 87.6 °C for narrow-leaved ragwort, and 134.0 °C for giant hogweed.

### 2.3. Hot Moisture Treatments

#### 2.3.1. Hot Steam and Hot Water

In contrast to the heat treatments, the exposure to hot steam and hot water led to fast killing of the seeds of all three species (Figure 5a–c, Appendix A). Particularly with hot steam, the most robust seeds were those of narrow-leaved ragwort. After 30 s, we found no decline in the survival rate of dry seeds as 100% of them germinated. A huge decrease in viability approx. 73% was observed after 1 min but even after 10 min there was still 1 viable seed left. Wet seeds of narrow-leaved ragwort were more vulnerable to the hot steam treatment as all of them died off after 5 min of exposure.

With giant hogweed, particularly wet seeds were very susceptible to hot steam as none of them survived even a treatment of 30 s. In contrast, 56.1% of the dry seeds were able to survive the exposure to hot steam for 1 min. After 5 min of treatment, all dry seeds of giant hogweed were eliminated. None of the giant hogweed seeds nor the seeds of narrow-leaved ragwort survived the treatment with hot water independent of the treatment time and moisture state. 

These results are in contrast to common ragweed, as dry and wet seeds of this IAP even survived hot water treatment. After 30 s imbibition in hot water, 2% of the wet seeds were still viable. In total, 5% of the dry seeds even survived the exposure to hot water for 1 min. The treatment with hot steam for 30 s did not have any effect on the seed viability of common ragweed, irrespective of them being dry or wet. However, after 1 min of exposure to hot steam, the survival rate decreased to 10.3% (dry) and 2% (wet), respectively. After 5 min of treatment with hot steam, all of the seeds were dead.

#### 2.3.2. Hot Foam

During the foam application, heat stability measurements were made with a laboratory thermometer using a penetration probe. The air temperature was 21.4 °C and conditions were sunny. The temperature achieved underneath the foam was 90.8 °C during the application. After 1 min, this initial temperature was reduced by only 7.7% to 83.7 °C. After 3 min, the temperature of the foam layer was still 67.8 °C, and 5 min after starting the application, the temperature decreased to 48.6 °C. However, as shown in Figure 6a–c (Appendix A), seeds of the three species reacted differently. The most resilient seeds were those of common ragweed. Only wet unprotected seeds were 100% killed. In the case of dry seeds, 75% survived even if they were just lying on the soil surface (unprotected). Wet seeds covered/protected by surrounding vegetation and leaf litter lost 56.5% of their viability. When the seeds were buried in 1 cm soil depth (fully protected), 39.5% were still viable after the treatment. Dry seeds were even more robust; 88% survived the treatment when covered/protected by vegetation and litter, and 75.3% of the seeds buried in soil were still viable. In contrast, seeds of narrow-leaved ragwort only survived the treatment with hot foam when they were buried in soil, irrespective of their moisture state. A similar result was observed with giant hogweed seeds. For dry seeds, the vegetation cover provided sufficient protection, leading to a survival rate of 31%.

## 3. Discussion

The results of our study revealed that seeds of prominent IAPs like common ragweed, narrow-leaved ragwort, and giant hogweed are relatively robust against heating depending on the details of the control process. For an effective eradication, temperatures of 80 °C and more were necessary to kill off wet seeds of all three investigated species. When seeds were dry and protected even an exposure to 100 °C for 48 h was not enough to fully eradicate them. Studies on the heat resistance of the seeds of these IAPs are scarce. To the best of our knowledge, the only scientific study on the heat resistance of common ragweed was performed by [31]. The authors demonstrated that unprotected wet seeds of common ragweed died off completely when they were exposed to 60 °C for 6 h. Even though similar treatment and analyzing methods were used, results deviate significantly from our findings. Another study was introduced by [32] testing the heat resistance of seeds of the related species giant ragweed (*Ambrosia trifida*). The author used microwave treatments for 10 and 15 min and demonstrated that all of the seeds of the species were killed, irrespective of the microwave power (24–150 °C). However, the comparability of results is limited due to different treatment methods and plant species. In addition, a few agricultural reports and recommendations are available which at least point out the possible heat tolerance of ragweed seeds with a view to avoid the composting of plants with already developed seeds [33].

In contrast, studies on the heat resistance of seeds of narrow-leaved ragwort are not available at all. Particularly, dry seeds proved themselves as very robust as 22% of the unprotected seeds survived a heat treatment with 100 °C for 48 h. When they were incorporated in soil, even 39% of the seeds were viable after this treatment. The plant originally derives from the South African Highveld, a plateau 1500 m above sea level covered by grassland and interspersed thorn trees, which is subject to strongly fluctuating climatic conditions like heavy rain, frost and long periods of drought which regularly causes wildfires (exposure to extremely high temperature) [34]. In this context, ref. [35] demonstrated that the germinative response of seeds of plants which are adapted to regular wildfires is correlated with their life cycle. This could be an explanation for the differences in the viability of common ragweed and narrow-leaved ragwort in surviving high temperatures. Particularly, dry seeds of narrow-leaved ragwort were more robust than those of common ragweed even though [36] showed that the achenes of common ragweed have a thick seed coat which could be a more efficient protection against added moisture and temperature than those of ragwort species [19]. Similar results were obtained by [37] who showed considerable variation in the survival probability depending on temperature and on the seeds’ ability to remain hard seeded during treatment (impermeability).

As with the other two species, studies on the heat resistance of giant hogweed seeds are scarce and are solely related to treatments in biogas fermenters. Whereas hard-coated seeds like common ragweed or narrow-leaved ragwort are more thermoresistant, giant hogweed seeds can be classified into the group of species without hard-shelled seeds and a quick inactivation rate [38,39]. For example, ref. [40] demonstrated that seeds of giant hogweed were completely killed after 40 days in a mesophilic digester (37 °C). In this context, another study on common ragweed seeds in biogas fermenters showed that a deposition time of 10 days at 37 °C was sufficient to kill off all of the seeds [31], but only in batch reactors and not in CSTRs (Continuous Stirred-Tank Reactors, [37]). However, the use of biogas fermenters to kill weed seeds cannot be applied to decontaminating huge amounts of soils. Additionally, in many European countries, common ragweed and narrow-leaved ragwort are mainly abundant and distributed along the road systems, but green cuttings of road verges and embankments are usually not allowed to be deposited in biogas fermenters within the European Union due to undesired waste additives like tyre abrasions. Thus, this biomass is usually removed from the road verges and embankments, composted and somehow recycled [41]. The maximum temperatures during professional composting is up to 70 °C which is not high enough to kill the seeds of our analyzed species.

In contrast to the heat air treatments particularly, the exposure to hot water caused a relatively quick mortification of seeds. Whereas 3–6% of the seeds of common ragweed survived a treatment for 1 min, seeds of narrow-leaved ragwort and giant hogweed died off immediately after 30 s of treatment. In addition, it was shown that hot foam treatment was not sufficient enough to cause a full eradication of seeds, particularly when covered by soil or vegetation. To the best of our knowledge, studies on the heat resistance of seeds exposed to hot foam are not available, but a couple of authors investigated the mortality of seeds exposed to hot moisture treatments. Authors of [42] investigated the seed survival of six different weed species (Poaceae, Brassicaceae, Portulacaceae, Amaranthaceae, Solanaceae, and Asteraceae) which were exposed to temperatures between 39 and 70 °C in water baths for up to 28 days. The authors showed that all of the seeds exposed to 70 °C died off completely after an exposure time of approximately 10 to 40 min. Authors of [43] demonstrated that seeds of giant hogweed were completely killed after 2 days exposure to 42 °C.

## 4. Materials and Methods

### 4.1. Plant Material

The dispersal units of common ragweed and narrow-leaved ragwort consist of a durable involucrum covering a hard-coated fruit (=achene s.str.); the single seed (morphological term) inside the achene is soft and comprises a well-developed embryo. The fruits of giant hogweed are referred to as two-winged, oval, paper-thin mericarps which contain one seed. For simplicity, the term “seed” will be used in the subsequent text to describe the dispersal units of the three species. When seeds of common ragweed mature in autumn, they drop off the plants finally. The seeds acquire dormancy during the dispersal process, and moist chilling is needed to overcome this dormancy [44]). Such processes happen under natural conditions—during the winter season. Based on lab studies, several authors recommend chilling treatments of ≥6 weeks at 4 °C to maximize the germination rate [45,46,47]. The fruits (=mericarps) of giant hogweed are one-seeded and can be easily ejected by wind, landing insects, or passing animals in what is the normal initiation of the spreading process [22,23,48]. Seeds of giant hogweed require cold, wet stratification for at least 6–8 weeks to break dormancy [49]. Nevertheless, the seeds are germinable even in autumn. In contrast, seeds of narrow-leaved ragwort collected in early autumn do not require any environmental stimuli for breaking dormancy that is acquired directly before release and hold for about one month [18,50].

Mature, dry seeds of each species were collected from mother plants at three different sites to avoid bias due to site-specific conditions during plant growth and seed ripening and maternal effects (Table 3). Immediately after collection, the seeds of all species were air purified, dried at room temperature, and placed at 4 °C on moist sand in darkness for at least eight weeks until the beginning of the treatments. These conditions guaranteed breaking dormancy at the start of the experiment.

### 4.2. Treatments

Temperature was applied in different ways (Table 4). The primary focus was on the hot air treatment in the lab. The hot moisture treatments were differentiated into hot water application and hot steam application. Finally, a field experiment was established to test for the efficacy of a hot foam application [51,52].

### 4.3. Hot Air Treatment

Hot air treatment was dry air heating of the seeds, applied in a climate chamber at 60, 70, 80, 90, and 100 °C for durations of 0.5, 1, 3, 6. 12, 24, and 48 h. Seeds were used either in a dry or wet condition. Seeds were moisturized by wrapping them in wet kitchen paper for 24 h immediately prior to the thermal treatment so that the seeds were imbibed with water. Another factor we included was the protection state, thus seeds were exposed to the heat treatment either (a) unprotected (simply placed open on a paper strip) or (b) protected (incorporated in 10 g potting soil filled in small linen bags). Each factor combination (temperature, test duration, protection state, moisture state) was tested with 100 seeds. As an unheated control, we included 100 wet and 100 dry seeds.

### 4.4. Hot Moisture Treatments

Dry or wet seeds of the three target species were steamed or put in hot water for 0.5, 1, 5, 10, 20, and 40 min at 90 °C. The steam and hot water treatments were performed using a laboratory steamer (Gester GT-D21B-2, Gester Instruments, Fujian, China). Additionally, a hot foam treatment was performed on wet and dry seeds in the field, using the hot foam system HWS 18 (IproTech, Iserlohn, Germany). In this experiment, we used the same seed samples as described in Table 1. Wet or dry seeds were treated (1) unprotected on the soil surface, or (2) protected by soil positioned 1 cm underneath the soil surface, or (3) partly covered by the surrounding vegetation which usually superimposes seeds when applying the hot foam spreading unit. Prior to this treatment, we performed heat measurements to determine the heat stability of the foam over time. As with the heat treatment, all factor combinations were tested with 100 seeds of each species (Table 4).

**Table 4 plants-13-00341-t004:** Treatment temperature, treatment time, protection state and moisture state of the heat treatment and hot moisture treatments of the seeds of common ragweed, narrow-leaved ragwort and giant hogweed; *n* = 100 seeds per treatment temperature, treatment time, protection state and moisture state, incl. 100 untreated control seeds per moisture state per treatment experiment.

Heat Treatment	
Treatment temperature:	60/70/80/90/100 °C
Treatment time:	0.5/1/3/6/12/24/48 h
Protection state:	unprotected (air heated) protected (incorporated in soil)
Moisture state of seeds:	dry wet
**Hot water and Hot steam**	
Treatment temperature:	90 °C
Treatment time:	0.5/1/5/10/20/40 min
Moisture state of seeds:	dry wet
**Hot foam**	
Initial temperature:	90.8 °C
Protection state:	unprotected on soil surface protected in 1 cm soil depth protected—covered by surrounding vegetation
Moisture state of seeds:	dry wet

### 4.5. Viability Testing

Seed survival was estimated by testing for viability in two steps. In the first step we tested for germination followed by a second step where we tested the viability of the seeds that did not germinate for living embryos. After the various heat treatments, the unprotected, pure heat-treated seeds as well as the seeds of the hot moisture treatments were put in Petri dishes on filter paper, saturated with tap water (20 seeds per dish) and moved to a climate chamber. Seeds from the field experiment that had been incorporated in soil (protected) were put into square Petri dishes (L × W × H: 120 mm × 120 mm × 17 mm) together with the surrounding original soil. Then, we added 15 mL of tap water to every Petri dish and transferred them to the climate chamber. 

The conditions for ragweed germination were set to 12 h full light at 25 °C and 12 h of darkness at 15 °C [47,53,54]. Narrow-leaved ragwort was exposed to 10 h of full light at 20 °C and 14 h of darkness at 10 °C (according to our own unpublished pre-trials). For giant hogweed, refs. [49,55] found that the optimum germination temperature is 10 °C without any influence of the light regime which was set to 10 h of full light and 14 h of darkness.

Petri dishes were checked every second day and were continuously kept wet during the germination test. In the pre-trials it became obvious that seeds that did not fully lose viability could develop a small radicle but did not show any further growth and finally decay. In avoiding such bias in the results, seeds were recorded as germinated when the radicle plus cotyledons were visible which ensured that seedlings developed further after the treatment.

Seeds which did not germinate within the 30-days period were consecutively tested for living tissue by use of 2,3,5-triphenyltetrazolium chloride (TTC) following the protocol of [53]. Seeds which were visibly intact but turned out to be empty when opened (no embryo developed) for the TTC tests were excluded from the results as the viability of these seeds was not affected by the heat treatments. Consequently, survival rates were calculated on the basis of the actuall embryo bearing seeds which were between 98 and 100 seeds per factor combination.

### 4.6. Data Analysis

Data analysis was performed using software R, Version 4.0.5 [56]. Basic data (number of viable seeds) were transformed to percentage of viable seeds in relation to all of the seeds tested in each sample (called “survival rates” in the text). Prior to statistical analysis, data exploration (collinearity and dispersion of response variable) was executed following [57]. Normal distribution of data and homoscedasticity were tested using Shapiro–Wilk and Levene’s tests, respectively. If the data were not normally distributed, a Kruskal-Wallis ANOVA on ranks was performed to check for differences. If homogeneity of variances was not given, a statistical analysis was executed using Welsh’s unequal variances *t*-test. Multiple regression analysis was performed using R-package lme4 [57] to check the impact of the explanatory variables on the results. For fine-tuning results of multiple regression analysis, generalized linear mixed models (GLMM) were constructed using R packages lme4 [58] MuMin [59], and AICcmodavg [60] aiming to detect the best model (factor combination) that can explain the survival rate of seeds of the three different species. Collinearity of the explanatory (fixed) variables “treatment temperature”, “treatment time”, “moisture state”, and “protection state” was tested using the R-package corrplot [61] and could be precluded. Random effects were included as treated seeds had to be separated into more than one Petri dishes (lack of space for 100 seeds in one Petri dish).

Models were selected by comparing the second order Akaike information criterion value corrected for small sample sizes (AICc). To identify the most parsimonious model based on the lowest AICc, we computed the AICc differences (ΔAICc) between the different candidate models. As a rough rule [62] proposed that models for which Δi ≥ 2 receive substantial support as the chance of the smaller AICc value being correct is approx. 73%. Sigma Plot, Version 14.5 (Systat Software, 2022) was used for graphical visualization of the results.

Seed survival curves were fitted based on the survival rates of the seeds under the given protection and moisture state using the R-package drc for dose–response curve fittings that include different adjustments, including Weibull models that can account for asymmetrical losses’ responses [63]. Again, the AICc values were compared to detect the most parsimonious model, based on the lowest AICc value. Authors of [63] proposed that models for which Δi ≥ 10 receive substantial support. Fitted equations were used to calculate the temperature required to reduce survival rate by 50% (ED50) and 99% (ED99). 

## 5. Conclusions

An effective heat treatment of seeds of IAPs could play a vital role in the deceleration of IAPs distribution and would be of major importance not only for road maintenance offices but also for landscape gardeners, farmers, and other professionals. As our laboratory experiment revealed, the seeds of noxious IAPs like common ragweed, narrow-leaved ragwort or giant hogweed are very robust and can withstand even temperatures of 90–100 °C for several hours. In contrast, hot water and hot steam showed a higher efficacy in the killing of seeds. However, under practical condition an extensive air or moisture heat treatment of biomass infested with seeds of IAPs or contaminated soil material is not feasibly due to time, effort and cost. In this context, the preventive management of these species by an efficient seed control could be substantial to avoid their further dispersal and increasingly expensive control measures.

## Figures and Tables

**Figure 1 plants-13-00341-f001:**
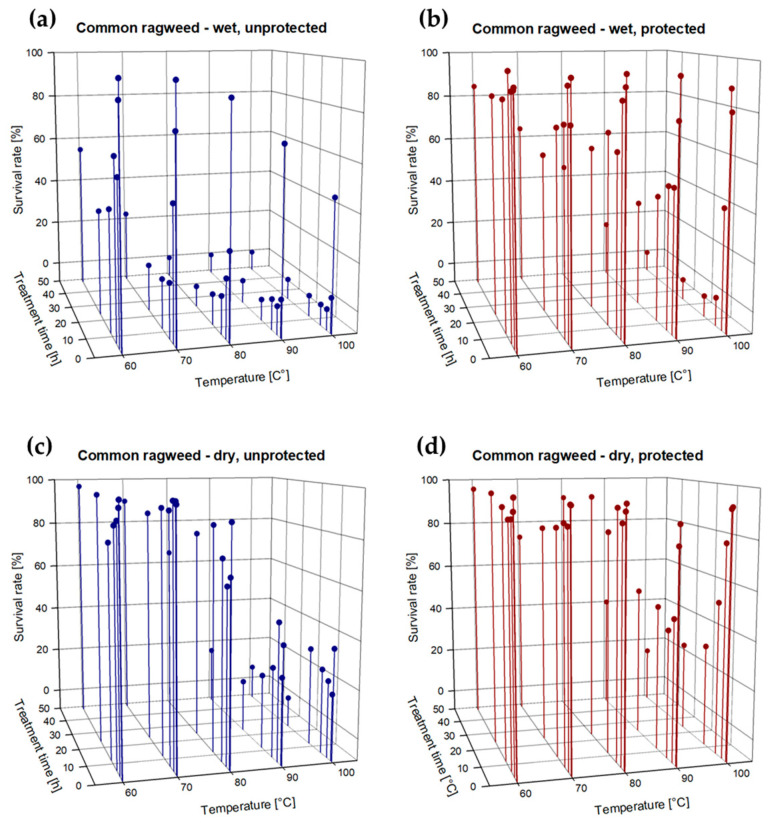
(**a**–**d**): Survival rate (%; dots) of (**a**) wet and unprotected, (**b**) wet and protected, (**c**) dry and unprotected, and (**d**) dry and protected seeds of common ragweed after exposure to different temperatures (60, 70, 80, 90, 100 °C) for 0.5, 1, 3, 6, 12, 24, and 48 h; *n* = 100 seeds per combination of factors: treatment time, temperature, moisture state and protection state = 14,176 embryo bearing seeds.

**Figure 2 plants-13-00341-f002:**
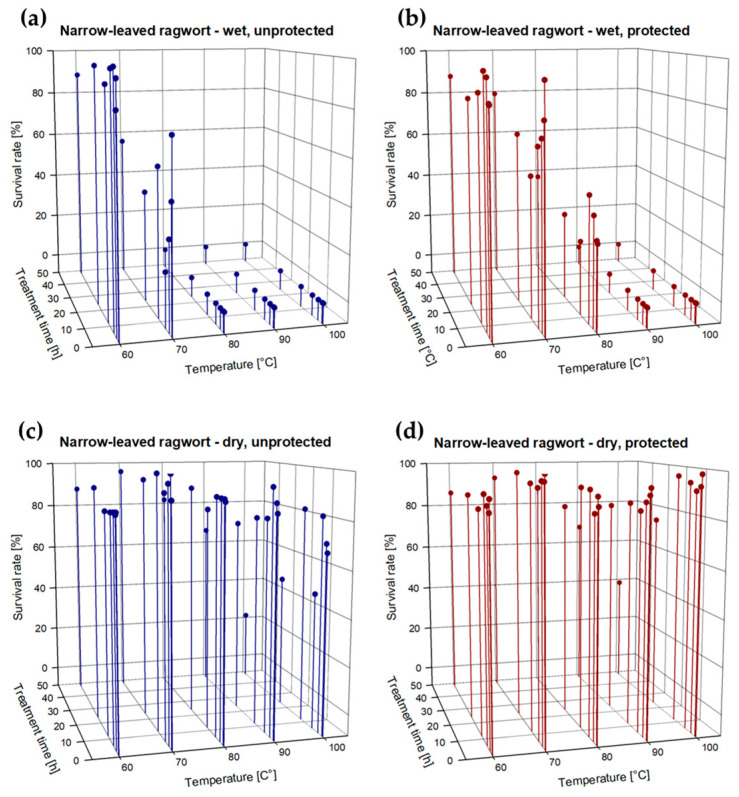
(**a**–**d**): Survival rate (%; dots) of (**a**) wet and unprotected, (**b**) wet and protected, (**c**) dry and unprotected, and (**d**) dry and protected seeds of narrow-leaved ragwort after exposure to different treatment temperatures (60, 70, 80, 90, 100 °C) for 0.5, 1, 3, 6, 12, 24, and 48 h; *n* = 14,160 embryo bearing seeds.

**Figure 3 plants-13-00341-f003:**
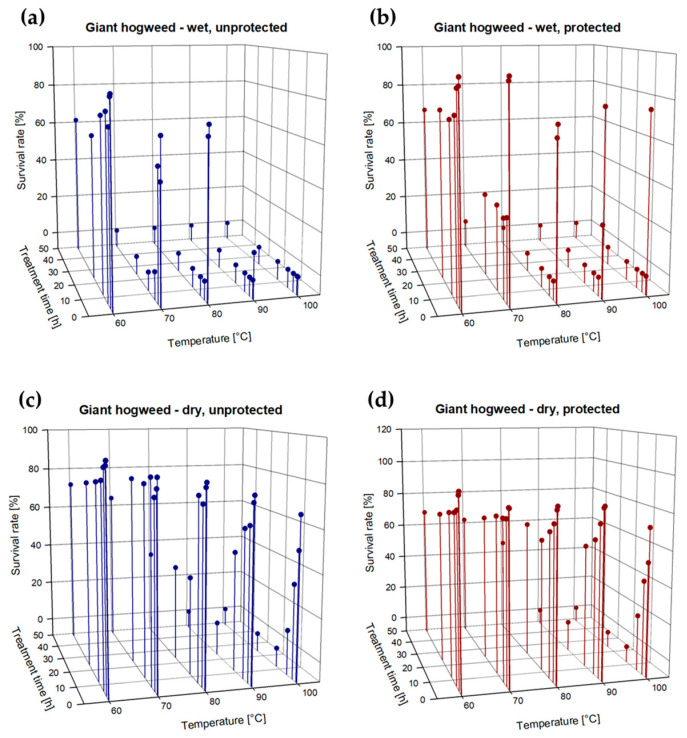
(**a**–**d**): Survival rate (%, dots) of (**a**) wet and unprotected, (**b**) wet and protected, (**c**) dry and unprotected, and (**d**) dry and protected seeds of giant hogweed after exposure to different treatment temperatures (60, 70, 80, 90, 100 °C) for 0.5, 1, 3, 6, 12, 24, and 48 h; *n* = 14,121 embryo bearing seeds.

**Figure 4 plants-13-00341-f004:**
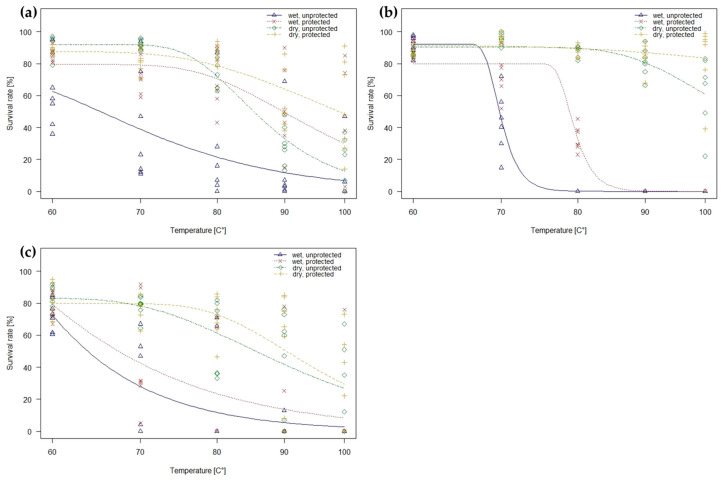
(**a**–**c**): Calculated curves of seed survival rates (%) of (**a**) common ragweed, (**b**) narrow-leaved ragwort, and (**c**) giant hogweed seeds (dry or wet, and unprotected or protected) after exposure to treatment temperatures of 60, 70, 80, 90, and 100 °C; logistic regression based on the most parsimonious model with Weibull probability distribution.

**Figure 5 plants-13-00341-f005:**
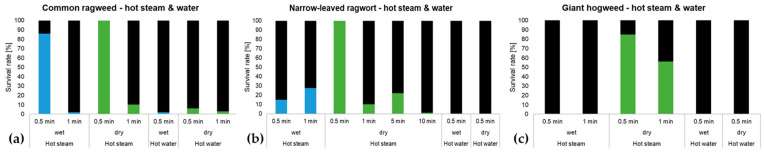
(**a**–**c**): Survival rate of wet (blue bars) and dry (green bars) seeds of (**a**) common ragweed, (**b**) narrow-leaved ragwort, and (**c**) giant hogweed after treatment with hot steam and hot water (both at 90 °C) for different treatment times, *n* = 2400 seeds per species; 100 seeds per factor combination; black proportion of the bars equals the mortality rate; results are only shown when there were still some viable seeds left.

**Figure 6 plants-13-00341-f006:**
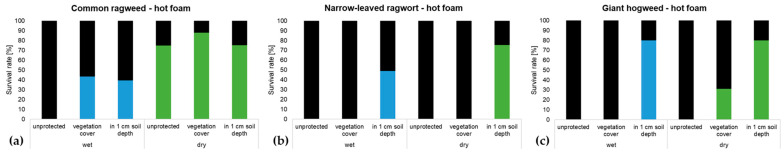
(**a**–**c**): Survival rate of wet (blue bars) and dry (green bars) seeds of (**a**) common ragweed, (**b**) narrow-leaved ragwort, and (**c**) giant hogweed after treatment with hot foam for 5 min unprotected, lying on the soil surface, covered with surrounding vegetation or buried 1 cm in soil; black proportion of the bars equals the mortality rate; *n* = 600 seeds per species; 100 seeds per factor combination.

**Table 1 plants-13-00341-t001:** Results of multiple regression analysis for the survival rate of seeds of common ragweed, narrow-leaved ragwort, and giant hogweed in relation to the explanatory factors treatment temperature, treatment time, moisture state and protection state; correlation coefficient R and regression coefficient R^2^ are related to the overall explanatory power of the analysis, including all factors; significance levels: *** *p* < 0.001, n.s. = not significant.

Species	Common Ragweed	Narrow-Leaved Ragwort	Giant Hogweed
Factors			
Treatment temperature	*p* < 0.001 ***	*p* < 0.001 ***	*p* < 0.001 ***
Treatment time	*p* < 0.001 ***	*p* = 0.126 n.s.	*p* < 0.001 ***
Moisture state	*p* < 0.001 ***	*p* < 0.001 ***	*p* < 0.001 ***
Protection state	*p* < 0.001 ***	*p* = 0.120 n.s.	*p* = 0.381 n.s.
correlation coefficient ©	0.83	0.86	0.84
regression coefficient (R^2^)	0.68	0.74	0.71

**Table 2 plants-13-00341-t002:** Summary of AICc values used for model selection of dependent variable seed survival rates of common ragweed, narrow-leaved ragwort and giant hogweed; number of estimated explanatory parameters and parameter combinations = 12; AICc = second order Akaike information criterion; ΔAICc = difference between AICc to the next most parsimonious model; R^2^ = proportion of variance explained by the factors.

	Explanatory Model	AICc	ΔAICc	R^2^
Survival Rate				
**Common ragweed**	Temperature (60, 70, 80, 90, 100 °C)	1297.2		0.53
	Treatment time (0.5, 1, 3, 6, 12, 24, 48 h)	1361.0		0.27
	Moisture state (wet vs. dry)	1348.9		0.29
	Protection state (unprotected vs. protected)	1369.6		0.19
*Model selection*	Temperature + Moisture State + Treatment time	1238.4	0.0	0.70
	Temperature × Moisture State + Treatment time	1241.1	2.7	0.68
**Narrow-leaved ragwort**	Temperature (60, 70, 80, 90, 100 °C)	1382.3		0.28
	Treatment time (0.5, 1, 3, 6, 12, 24, 48 h)	1424.7		0.02
	Moisture state (wet vs. dry)	1332.2		0.47
	Protection state (unprotected vs. protected)	1421.9		0.01
*Model selection*	Temperature × Moisture state	1151.9	0.0	0.86
	Temperature × Moisture State + Treatment time	1152.8	0.9	0.87
	Temperature + Moisture State	1240.9	88.1	0.74
**Giant hogweed**	Temperature (60, 70, 80, 90, 100 °C)	1340.4		0.38
	Treatment time (0.5, 1, 3, 6, 12, 24, 48 h)	1386.5		0.14
	Moisture state (wet vs. dry)	1368.7		0.20
	Protection state (unprotected vs. protected)	1401.9		0.01
*Model selection*	Temperature + Moisture State + Treatment time	1241.9	0.0	0.71
	Temperature × Moisture State + Treatment time	1244.1	2.2	0.70

**Table 3 plants-13-00341-t003:** Origin, coordinates (WGS84), habitat type, and harvest date of seeds of common ragweed, narrow-leaved ragwort, and giant hogweed.

Species	Origin	Coordinates (N/E)	Habitat Tye	Harvest Date
**Common ragweed**	Augsburg	48°22′32.0/10°50′19.3	Arable field	14 October
	Mühldorf am Inn	48°16′11.1/23°32′57.6	Federal road verge	13 October
	Bernau am Chiemsee	47°49′02.8/12°22′17.7	Highway verge	15 October
**Narrow-leaved ragwort**	Munich	48°16′34.5/11°38′31.0	Highway parking	8 September
	Meitingen	48°32′30.0/10°50′31.4	Federal road verge	2 September
	Vienna	48°10′54.9/16°22′54.8	Railway station	5 October
**Giant hogweed**	Emersacker	48°29′37.2/10°39′43.2	Logging road	13 September
	Köfering	49°23′55.2/11°50′33.6	Highway embankment	20 September
	Viereth-Trunstadt	49°55′59.0/10°46′55.6	Commuter parking lot	7 September

## Data Availability

The data presented in this study are available on request from the corresponding author.

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
