# Peer review of "Heat Treatment of Seeds to Control Invasive Common Ragweed (*Ambrosia artemisiifolia*), Narrow-Leaved Ragwort (*Senecio inaequidens*) and Giant Hogweed (*Heracleum mantegazzianum*)"

_plants, 2024, doi:10.3390/plants13030341_

Round 1
Reviewer 1 Report
Comments and Suggestions for Authors
Manuscript title:
Heat treatment of seeds to control invasive common ragweed (Ambrosia artemisiifolia), narrow-leaved ragwort (Senecio inaequidens) and giant hogweed (Heracleum mantegazzianum)
The manuscript is particularly strong regarding the less studied topic and the experimental setup on the seeds treatment and management of some European invasive species ……. The manuscript regarding the topic and results presented is of interest to plant science community and revisions based on the comments below are recommended before considering for publication.
Major comments
· Insufficient Abstract: In the abstract, it would be even better to have a sentence as a future perspective.
· Line 85-90, the aim or hypothesis of the study is not clear, and the approach is missing ….
· Lake of scientific literature to support the statements and findings throughout the manuscript…... I have made some suggestions for that and more need it….
· More information is needed for ALL TABLE captions and define the abbreviations and units that are used. And adjust the significant figures for the table and manuscript.
· I have a major concern about the results and discussion section. The authors describe the results and compare the results with previous studies, however, insight mechanisms are still insufficient.
Detailed comments:
Introduction:
Line 41-46: A complicated sentence, please revise and check the grammar
Line 50: A reference is needed here, for example, you can use:
https://doi.org/10.3390/agronomy12020423
Line 63-69: A complicated sentence, please revise
Line 70: A reference is needed here, for example, you can use: https://doi.org/10.1016/j.scitotenv.2020.142822
R&D section
Figure 1-3, even through the effect is clear from the discussion, however, it would better to highlight the significant effect on he graphs too, to help the readers
These sections repeat information already presented and explain things in an unnecessarily complicated way. The quality of the manuscript would benefit from the whole section being condensed, Line 141-161, Line 217-236, Line 287-318…..
MM section
Literature references are missing for all sub-sections. It would be better to cite the references that the procedure adopted.
Additional info is needed for the table caption, most importantly significant figures.
In the MM section, what is the quality control (QC) data? There is no mention of the QC.
In general, how many times you’ve recorded the data,? duplicate? Triplicate?..... what you mentioned in the text is not clear, please elaborate more on this
Conclusion
The section should not be a summary of your study or an extension of the discussion. This section should illustrate the mechanistic links of the findings of this study. The conclusions should answer the hypothesis of your study and should focus on the implication of your findings. Remember that the conclusions must be self-explanatory. This section should still highlight the novelty and implication of your study also.
Comments on the Quality of English Language· Minor grammar and punctuation issues need to be addressed.
Author Response
Thank you very much for reviewing our manuscript which clearly contributed to the improvement of our manuscript. Unfortunately, not all comments were clear so please find our questions and justifications in the word.file enclosed.
Best regards,
Rea Maria Hall

Reviewer 2 Report
Comments and Suggestions for Authors
The paper is interesting, scientifically well organized, and contributes to the control of some invasive alien species.
Just two methodological comments
- if survival rates were calculated based on seeds carrying the viable embryo even after germination tests, it is unclear why these tests were carried out. Was it not enough to use the TTC test directly on the seeds after heat treatment and exclude the germination test?
-There is no mention of how to control the growth of fungi in Petri dishes during germination tests.
Some minor observations are reported in the attached pdf.

Author Response
Thank you very much for reviewing our manuscript which clearly contributed to the improvement of our manuscript. Please find our comments and justifications in the word.file enclosed.
Best regards,
Rea Maria Hall

Reviewer 3 Report
Comments and Suggestions for Authors
Authors address a very important topic from the point of view of the invasion of alien species with effects on biodiversity and the economy and human health. I find that there are some deficiencies in the preparation of the manuscript that I believe are easy to correct, however it is imperative to correct them so that the manuscript can be published.
I list them below:
1) the effect of the combinations was evaluated through germination? 2) How many combinations were there in total? Please explicitly state the experimental design for this experiment. 3) If the effect was evaluated through germination, were replicas of 100 seeds each placed? Or how were the 100 seeds handled statistically, considering that the germination response is binomial for each seed?
Please explicitly indicate how many replicates were used for each treatment combination.
The conclusions are not based on the results. These are again the arguments used in the introduction to justify the work. It would be important to indicate which combination of treatments decreased seed survival in each species. And indicate if it was the same for all of them. How viable would they be to apply in practical matters.
I also made some specific comments about the PDF

Author Response

(The authors gave the same response as above.)

Reviewer 4 Report
Comments and Suggestions for Authors
The aim of this study is to find a potential solution for an important problem, the spread of invasive alien plant species. While the results of this study can have practical importance and will be interesting to the readers, the text must be improved and some of the methods and results must be clarified.
The Introduction part is generally fine, but the references in the text must be checked. For example, in Ln 350-351 reference [18] is used in the text where dormancy of Senecio seeds is described, but I could not find the information about seed dormancy of this species in the cited paper. Similarly, references [22, 36] do not seem to contain information on the dormancy breaking requirements of the giant hogweed. References 46 and 47 (Ln 364) do not seem to apply to the text where they appear. The order of the references in the text does not correspond to the list of references (e.g. reference to the R is 51 in the text and 49 in the list).
In the description of results, it is stated that 98% of untreated seeds germinated. What is referred to as “dry untreated” and “wet untreated” seeds? All germination tests require imbibition of the seeds. Does it follow that there were two groups of seeds used as a control, but these seeds were treated in the same way? Moreover, the authors write (Ln 355) that seeds were subjected to cold stratification, which, by definition, involves seed imbibition (cold treatment is applied to wet seeds). Otherwise, this is just cold storage and is unlikely to break seed dormancy. If seeds were wet during stratification, were they dried afterwards, before the dry seed treatments?
It looks like the control treatments are not shown in the figures (1-3). If this is correct, the results of germination and viability tests with the untreated (control) seeds must be stated in the text or shown otherwise. Names of the species must be included in the figure captions.
I should advise not to use “germination rate” in the sense of percentage of germinated seeds (Ln 95).
Could the authors argue that it is appropriate to model the effect of temperature on seed viability loss while not taking into account the duration of the treatment? I am not confident that it is correct. What is the practical value of this estimation, if e.g. ED99 for giant hogweed is 95.4 °C without any specification of the treatment time?
Ln 100: in this paragraph the reference should be to Figure 1 c, d. Further in the text (Ln 108) a reference to Figure 1 a, b is necessary. It would be better to reorder the panels in the figure according to the text. The same applies to Figures 2 and 3.
Ln 120 (Figure 1 caption) Is “n” the total number of seeds used? I would be more useful to indicate the number of replicates (in this case, apparently, 100 seeds. The same applies to Figures 2 and 3.
In Figures 5 and 6 the stacked bars are not necessary, especially as it is not explained what the black portion of the bar is.
The paragraph “2.3.1. Hot steam & Hot water” is difficult to read and could be restructured.
Concerning the methods used: it is stated in Ln 396 that there were 20 seeds per Petri dish. How many Petri dishes (replicates) were there? If all 100 seeds were tested, it would follow, that there were 5 replicates, but this must be clearly stated.
Please specify what were the fixed and random effects in the GLMM (Ln 431-432).
It would be good to specify which data were used for fitting the seed survival curves (Ln 443).
The Conclusions should be more concise and focused on the results, not the general information that is included in the Introduction. All the sentences that include citations should be moved to the Discussion.
Comments on the Quality of English LanguagePlease find some suggestions below (this is not the exhaustive list of corrections that may be required, especially in grammar and syntax).
Introduction
Ln 45-46: I suggest rewriting this sentence, “dispersal in space” and “dispersal in time” may be misleading. Dispersal is, by definition, distribution in a geographical area. “Dispersal in time” – change to “seed survival”.
Ln 58: form persistent soil seed bank?
Ln 74: germination during interim storage would reduce the number of viable seeds in the soil (presumably, the seedlings would not survive).
Ln 78: herbicides? (plural)
Ln 89-90: The last sentence could be re-written, e.g. “Based on the results of this study, the efficacy of weed control measures may be enhanced.”
Results
Ln 95: “I” is a typo or a mistake?
Ln 135: replace “even” with “as many as”
Ln 146-147: “. Particularly, with dry seeds the same...” - this sentence is not clear.
Ln 167: survival of common ragweed seeds
Ln 167: “affected with high significancy” - significantly affected by...
Ln 182: “they” - the?
Ln 221: decrease in viability?
Discussion
Ln 270, 320: “mortification” - replace with “killing” or “viability loss”
Ln 271: “an more” - and more
Ln 294-295: From this sentence it is hard to understand what is the connection between germination and wildfires in this species.
Ln 322-323: the word order in this sentence is confusing
Methods
Ln 335: “involucrum” – I am not familiar with use of this term in the context of seed morphology, perhaps “pericarp” is the correct term?
Ln 337: mericarps (plural)
Ln 346: the same term (moist chilling) should be used, for consistency (the synonym of cold stratification).
Ln 351: holds? (referring to dormancy)
Ln 355: presumably, seeds were stratified on some moist substrate, this should be described.
Ln 366: does this sentence mean that the seeds were incubated in a climate chamber at 60-100 °C for 0.5-48 hours? It should be rewritten more clearly.
Ln 368: moistened
Ln 380: "Therefore" - in this experiment?
Ln 392: tested (past tense)
Ln 393-394: tested the viability of the seeds that did not germinate
Ln 408: "not fully devitalized seeds" could be replaced by "seeds that did not fully lose viability"
Ln 426-428: I suggest rephrasing; instead of "normality not given" - "if the data were not normally distributed”; "if the data were heteroscedastic”
Ln 444: it probably should be specified here (rather than in the results) what kind of Weibull model was fitted (the number of parameters).
Author Response

(The authors gave the same response as above.)

Round 2
Reviewer 1 Report
Comments and Suggestions for Authors
I am happy to see the manuscript improved nicely. The authors addressed all my comments adequately.
Author Response
Dear Reviewer,
thank you very much for your positive feedback and for your effort and contribution to the improvement of the manuscript.
Best regards,
Rea Maria Hall
Reviewer 4 Report
Comments and Suggestions for Authors
" Random effects were included as treated seeds had to be seperated 451 into more than one Petri dishes (lack of space for 100 seeds in one Petri dish)." - doest it mean that the number of the dish was the random effect in the model? This has to be stated.
Comments on the Quality of English LanguageLn 95 "wether protected" - either?
Author Response
Dear Reviewer,
thank you very much for your recommendations as well as your effort and contribution to improve our manuscript. You are completely right, the "wether" in ln 93 should be an "either". We changed that.
Furthermore, we want to admit, that in line 748 it is already stated that random effects were included due to random number of Petri dishes.
Best regards,
Rea Maria Hall